# Cellular Metabolic Regulation in the Differentiation and Function of Regulatory T Cells

**DOI:** 10.3390/cells8020188

**Published:** 2019-02-21

**Authors:** Ye Chen, Jacob Colello, Wael Jarjour, Song Guo Zheng

**Affiliations:** 1Division of Rheumatology and Immunology, Department of Internal Medicine at Ohio State University of Medicine and Wexner Medical Center, Columbus, OH 43201, USA; Ye.Chen@osumc.edu (Y.C.); Wael.Jarjour@osumc.edu (W.J.); 2Division of Rheumatology, Department of Medicine at Penn State College of Medicine, Hershey, PA 17033, USA; jcolello@pennstatehealth.psu.edu

**Keywords:** cell metabolism, T cells, Foxp3, mTOR

## Abstract

Regulatory T cells (Tregs) are essential for maintaining immune tolerance and preventing autoimmune and inflammatory diseases. The activity and function of Tregs are in large part determined by various intracellular metabolic processes. Recent findings have focused on how intracellular metabolism can shape the development, trafficking, and function of Tregs. In this review, we summarize and discuss current research that reveals how distinct metabolic pathways modulate Tregs differentiation, phenotype stabilization, and function. These advances highlight numerous opportunities to alter Tregs frequency and function in physiopathologic conditions via metabolic manipulation and have important translational implications.

## 1. Overview of the Effects of Cellular Metabolism on Tregs

Regulatory T cells (Tregs) are crucial for immune homeostasis and the control of inflammatory disorders [1,2,3,4]. We mainly focus the discussion on CD4+CD25+Foxp3+ Tregs in this review. Tregs subsets include: thymus derived Tregs (tTregs), peripheral derived Tregs (pTregs) which are generated extrathymically at peripheral sites, and iTregs that are induced ex vivo following TCR stimulation in the presence of transforming growth factor β [5]. Tregs-mediated protection has been applied in numerous preclinical models of autoimmune diseases and transplantation, which informs on their therapeutic potential for human diseases [6,7,8,9].

Like conventional CD4+ T cells, Tregs also have a high degree of plasticity related to different transcriptional programs [10,11,12,13], which are in turn impacted by cellular metabolism [14]. Recent findings show that Tregs use glycolysis and fatty acid oxidation differently than effector conventional T cells and naïve T cells [15]. Compared to effector conventional T cells, mouse Tregs oxidize lipids at higher rates and exhibit low glycolytic flux in vitro. On the other hand, the modest energy and biosynthesis demands of naïve T cells are typically met by the tricarboxylic acid cycle, lipid oxidation, and glycolysis (Figure 1) [16]. Moreover, glycolysis inhibits Tregs differentiation and promotes Tregs expansion, whereas fatty-acid oxidation (FAO) promotes Tregs differentiation [15,17]. In addition, the expression of Foxp3 in Tregs inhibits Myc expression and reduces glycolysis, which can be suitable for Tregs in low glucose condition [17]. Foxp3, in turn, induces oxidate phosphorylation and increases the ratio of oxidized nicotinamide adenine dinucleotide (NAD) over the reduced form (NADH), allowing Tregs to survive in elevated lactate environments [17]. Furthermore, autophagy is one of the first responses when cells experience nutrient limitation and is critical for Tregs fitness. Deficient autophagy leads to the upregulation of the mechanistic target of rapamycin (mTOR) and c-Myc (*Myc* proto-oncogene) expression as well as an increase in glycolysis, resulting in impaired Tregs function [18,19].

Different chain lengths of fatty acids have dissimilar effects on Tregs differentiation. Adding a short chain fatty acid to mouse or human naïve CD4+ T cells enhances Tregs differentiation, while a long chain fatty acid (LC-FA) decreases Tregs differentiation [20]. Our previous work demonstrated that sodium butyrate, which belongs to the short-chain fatty acid family, promotes Tregs induction and displays therapeutic potential in several inflammatory disorders [21]. However, Raud et al. recently reported that Carnitine palmitoyltransferase 1a (Cpt1a), a critical regulator of LC-FA oxidation, is largely dispensable for Tregs generation [22].

mTOR is a 289 kDa serine/threonine protein kinase that is highly evolutionarily conserved and has two complexes mTORC1 and mTORC2 [23]. It can directly affect T cell proliferation and differentiation through the integration of environmental cues such as energy stores, nutrients, and growth factors; and can be selectively inhibited by rapamycin [24]. Generally, mTORC1 is more sensitive to rapamycin than mTORC2 [25], however, in naïve CD4+ T cells, mTORC1 and mTORC2 have essentially the same sensitivity to rapamycin [26]. This review describes the effects of mTOR signaling dependent cellular metabolic regulation on Tregs phenotype and differentiation/suppressive function. Moreover, we discuss the role of mTOR in its modulation of T cell metabolism, which could provide targets for metabolic manipulation.

## 2. mTOR

As a member of phosphatidylinositol-3 kinases (PI3K) family, mTOR contains two N-terminal HEAT domains (binding domain), which are important for protein–protein interactions. It also includes an FRB region (rapamycin binding domain of mTOR), a FAT domain (a domain in PI3K-related kinases), a structurally supportive C-terminal FATC domain (a domain in PI3K-related kinases), and a kinase domain [27]. During T cell activation, T cell receptor (TCR) stimulates the mTORC1 and mTORC2 via triggering the recruitment of PI3K to the TCR receptor [28]. The activation of PI3K leads to activation of the serine–threonine kinase AKT (also known as protein kinase B) by pyruvate dehydrogenase kinase 1 (PDK1), following the activation of mTOR signaling [29]. Additionally, PI3K can directly induce the activation of mTORC2 [30]. Diverse environmental inputs can be integrated into the mTOR pathway. For example, through mTOR, metabolic cues and immune signals have an ability to direct T cell fate decisions [31]. Moreover, co-stimulatory signals, TCR and cytokine can activate mTOR via PI3K-AKT signaling to meet energy demands and activate T cells.

### 2.1. mTOR and Tregs Differentiation

The most profound function of mTOR in Tregs generation was first revealed using the selective inhibitor of mTOR, rapamycin, which decreased the production of effector T cells and increased the generation of Tregs [32]. Furthermore, a lack of mTORC1 signaling may lead to a failure of differentiation from naïve CD4+ T cells to Th17 lineage. When mTORC2 and mTORC1 were both mutually absent, however, naïve CD4+ T cells were differentiated into Foxp3+ Tregs [33]. This research underscores the significant role of mTOR as a fundamental regulatory factor in the differentiation of Tregs and Th17 cells (Figure 2).

### 2.2. mTOR and Tregs Function

Tregs generation is enhanced during an immune response by inhibition of mTOR. Such activity is considered a required step in maintaining Tregs suppressive capabilities. Recent evidence has revealed a critical role of mTORC1 complex in the development of Tregs suppressive activity [34] (Figure 2). If Raptor is specifically deleted from Tregs, mTORC1 is disrupted. This leads not only to a profound loss of Tregs suppressive activity, but also causes the development of a fatal early-onset inflammatory disorder. Mechanistically, cholesterol/lipid metabolism is enhanced through Raptor/mTORC1 signaling in Tregs [34]. In order to establish Tregs functional competency, the mevalonate pathway can up-regulate the Tregs suppressive molecules CTLA-4 and ICOS, as well as coordinate Tregs proliferation. Inhibition of the mTORC2 pathway is partly involved in maintaining Tregs function by mTORC1 [34]. Nevertheless, mTOR signaling is critical for properly programming activated Tregs function in order to protect tissue homeostasis and preserve immune tolerance. Tregs-specific deletion of mTOR impairs Tregs function and homeostasis, resulting in the spontaneous effector T cell activation and in the development of inflammation in barrier tissues, which is correlated with the reduction in the local tissues of both peripheral Tregs (pTregs) and thymic-derived Tregs (tTregs) [35]. In contrast, Toll-like receptor (TLR) signals enhance Tregs proliferation through mTORC1 signaling pathway, glucose transporter 1 (Glut1) upregulation, and glycolysis. However, these signals decrease the suppressive ability of Tregs [36]. It is likely that the TLR signal results in high levels of pro-inflammatory cytokines such as IL-6, IL-1, TNFα, and these pro-inflammatory cytokines decrease Tregs functionality even as Tregs maintain mTORC1 expression.

### 2.3. mTOR and Tregs Expansion

Although rapamycin is commonly used to block tumor cell growth, it is interesting that proliferation of CD4+ T cell and activation-induced cell death cannot be blocked by rapamycin in vitro and in kidney transplant rejection [37]. On the contrary, expanded Tregs may suppress the proliferation of effector T cells in vitro as well as prevent allograft rejection in vivo [37,38]. A new study has revealed that branched-chain amino acids (BCAAs) could be essential for the maintenance of Tregs profiling state, through metabolic reprogramming of the amino acid transporter solute carrier family 3 member 1 (Slc3a2) dependent pathway. Slc3a2-deficient Tregs impair the mTORC1 pathway and show lower proliferation ability [39] (Figure 2). The expansion of Tregs by rapamycin usually requires the addition of IL-2. Thus, even when mTORC1 is inhibited by rapamycin [40], IL-2 maintains the ability to expand Treg cells.

### 2.4. mTOR and Tregs Migration

As one of the complexes of mTOR, mTORC2 has been proven to have control of spatial aspects of cell growth via actin reorganization [41,42]. The immune-modulatory function is critically interrelated with the migration of activated Tregs to inflammatory tissue [43]. A recent study demonstrated that glycolysis was beneficial for their relocation. Migration of Tregs to inflamed tissue was initiated by pro-migratory stimuli through a PI3K-mTORC2-mediated signaling pathway, which culminated in stimulation of the enzyme glucokinase (GCK). Subsequently, GCK increases cytoskeletal rearrangements by interacting with actin. If Tregs lack this pathway, they will still be functionally suppressive but will fail to migrate to skin allografts as well as inhibit rejection [44] (Figure 2).

## 3. Promising Metabolic Targets to Manipulate Tregs Frequency and Function

mTOR is a very important regulator of cell survival and plays bidirectional roles in Tregs induction and function. The absence of mTOR signaling dramatically increases Tregs generation and inhibits the function of Tregs. To advance therapeutics and promote homeostasis of the immune system, it is necessary to identify more specific targets that modify Tregs function or induction. Next, we discuss the effects of some mTOR interrelated metabolic regulators on Tregs phenotype.

### 3.1. Hypoxia-Inducible Factor 1α (HIF1α)

The transcription factor hypoxia-inducible factor (HIF1α) is a necessary protein for sensing oxygen saturation and subsequently initiating the cellular response to hypoxia [45,46]. It also is closely associated with Tregs and the mTOR pathway. Furthermore, impeding glycolysis up-regulates production of Tregs that occurs through inhibition of mTOR-mediated induction of HIF1α [45]. Interestingly, HIF1α inhibits Tregs differentiation in a transcription-independent manner [47], while not affecting *Foxp3* mRNA levels. HIF1α exerts this inhibition through promoting the degradation of Foxp3 protein by ubiquitination (Figure 3). Moreover, lack of HIF1α promotes Tregs induction and protects mice from autoimmune neuro-inflammation [45].

Inflammation and hypoxia are two independent factors regulating the balance between Th17 and Tregs. Our group reported that tTregs are unstable in the inflammatory environment and fail to suppress collagen-induced arthritis [48]. Mechanistically, the presence of the inflammatory cytokine IL-6 converts tTregs to Th17-like cells in vitro [2,11,12,49], and IL-6 also increases HIF1α expression in a stat3-dependent manner [47]. Up to this date, it is unclear whether IL-6 regulates Tregs stability via metabolic alteration through HIF1α. A number of published articles reported that HIF1α is a key regulator in inflammation and autoimmune diseases, such as systemic lupus erythematosus (SLE) [50], rheumatoid arthritis (RA) [51,52], type 1 diabetes (T1DM) [53], multiple sclerosis (MS) [54], psoriasis [55], and inflammatory bowel disease (IBD) [56]. These findings indicate that HIF1α is a potential target to manipulate Tregs phenotype in autoimmune diseases.

### 3.2. AMP-Activated Protein Kinase (AMPK)

AMPK senses the cellular AMP/ATP ratio and is activated by low energy balance (high AMP/ATP ratio) [57,58,59]. Activated AMPK promotes FAO via upregulating a series of lipid oxidation related genes, such as Acetyl-CoA carboxylase 1 (ACC1), Acetyl-CoA carboxylase 2 (ACC2), Cpt1a, and sterol regulatory element binding transcription factor 1 (SREVP-1c) [60]. In addition, AMPK also regulates glycolysis via adjusting the expression of Glut1 [61]. AMPK is responsible for Tregs differentiation via regulating the balance of FAO and glycolysis. Activation of AMPK by metformin increases Tregs induction; and in murine models inhibits the progression of experimental autoimmune encephalomyelitis (EAE), and inflammatory bowel disease [15,62,63]. TCR activates both AMPK and mTOR, the latter active kinase is a negative regulator for the former one under limited nutrient condition [64]. Interestingly, the absence of AMPK has no effect on Tregs differentiation even though it enhances mTORC1 activity. Liver kinase B1 (LKB1) is a best-studied upstream kinase of AMPK and is an important metabolic sensor of Tregs. The absence of LKB1 in Foxp3+ Tregs limits the number and function of Tregs, and the effects of LKB1 on Tregs generation and function are independent of AMPK and mTOR [65,66]. Moreover, a recent paper demonstrated that Tregs differentiation was independent of the AMPK-Driven LC-FAO [22].

### 3.3. Leptin

Leptin is a cytokine-like hormone and is structurally similar to IL-6. Leptin mediates metabolism and T cell function [67,68]. Chronic leptin- and leptin-receptor deficiency is correlated with resistance to autoimmunity and high susceptibility to infection [69,70]. For example, leptin levels in SLE patients are correlated with regulatory T cell frequency [71]. De Rosa et al. reported that both leptin and its receptor are constitutively expressed in freshly isolated human Tregs. Increased leptin signaling acts as an antagonist during Tregs proliferation (Figure 3) [72].

### 3.4. Peroxisome Proliferator-Activated Receptors (PPARs)

Peroxisome proliferator-activated receptors (PPARs) are nuclear hormone receptors that function to regulate cell growth, homeostasis, and differentiation. PPARα, β/δ, and γ are three primary isoforms, each with distinct functions and tissue distribution [73]. PPARs play an important role in peroxisomal mediated β-oxidation of FAO. When PPARs are activated with specific ligands, conformational changes occur, resulting in heterodimerization with retinoid X receptor (RXRα), which then binds to promoter regions of target genes involved in FAO [74,75]. PPARs also regulate glucose metabolism. Recent studies have demonstrated that agonists of PPARs inhibit inflammatory and immune responses in non-alcoholic fatty liver disease, at least in part, through increased expression of Foxp3 and induction of Tregs (Figure 3) [76,77]. Although studies have revealed a clear picture of anti-inflammatory function by PPAR-γ, the action of other PPARs on the Tregs population remains uncertain. Recent studies have focused on the relationship between PPARs and visceral adipose tissue (VAT) Tregs. VAT Tregs are a unique subset of Tregs that uniquely express PPARγ and are specifically recruited to adipose tissue to suppress the inflammatory process [78]. In mice that specifically lack PPARγ in Tregs, VAT Treg cell population is reduced. These mice display enhanced insulin resistance and increased susceptibility to diabetic pathology [79,80]. These findings suggest that PPARγ may be a promising target for obesity-associated insulin resistance (IR). However, the absence of PPARγ in VAT Tregs does not perturb their frequency in aged mice, and VAT Tregs show a gene expression profile more similar to fat effector conventional T cells than splenic Tregs [81]. One possible reason is that long term inflammation in aged mice may change the characteristics of Tregs and promote the transdifferentiation. These results highlight the importance of PPARγ in Treg differentiation, migration, and function although aging and chronic inflammation may affect the role of PPARγ in Treg biology. 

### 3.5. The Aryl Hydrocarbon Receptor (AHR)

AHR exists as a receptor and transcription factor, which is essential for xenobiotic metabolism and shows a cital function in immunity [82]. AHR has a high-affinity ligand, TCDD (2,3,7,8-tetrachlorodibenzo-*p*-dioxin) and when activated in vivo, AHR–TCDD complex leads to the induction of CD4+CD25+Foxp3+ Tregs. Alternatively, 6-formylindolo [3,2-b] carbazole (FICZ) may activate AHR to interfere with Tregs differentiation, boosting Th17 cell differentiation and worsening experimental autoimmune encephalomyelitis (EAE). Therefore, AHR regulates the balance of Tregs and Th17 cell differentiation in a ligand-specific manner (Figure 3) [83] and can be a unique target for immunosuppression therapy.

### 3.6. Interleukin 2 (IL-2)

IL-2 was first found as a T cell growth factor and plays an important role in T cell proliferation and differentiation [84]. We confirmed, along with several groups, that IL-2 is still necessary to induce and expand Tregs [10,85]. Zeng et al. demonstrated that IL-2 enhancement of Tregs function was dependent on the activation of mTORC1 [34]. Interestingly, IL-2 may partner with rapamycin (mTOR inhibitor) in Tregs expansion in vitro (Box 1, Figure 3). Low dose IL-2 is a promising method for the treatment of autoimmune diseases like lupus [86,87], T1DM [88], and graft-versus-host (GVHD) [89,90].

## 4. Concluding Remarks and Perspectives

Recently, research has highlighted the complex roles of intrinsic metabolic pathways in the development and function of Tregs, which may have significant implications on immune diseases and responses. In this review, we briefly summarize the roles of metabolic sensors in the biological features of Tregs as well as their potential to be targets of clinical immune-modifying therapies (Figure 3). Immunometabolism is a promising new field, and profound questions remain to be answered (Box 1). 

Box 1Some unsolved questions in the metabolic regulation of Tregs.
The different roles of mTORC1 and mTORC2 in Treg induction, migration, expansion, and function;The different metabolic profiles of Tregs during steady states and inflammatory conditions;Identification of metabolic factors that correlate Tregs development, function, and expansion with environment cues;The basis of the requirement of high doses of IL-2 (mTOR activator) along with rapamycin (inhibit mTOR) to expand Tregs in vitro.


## Figures and Tables

**Figure 1 cells-08-00188-f001:**
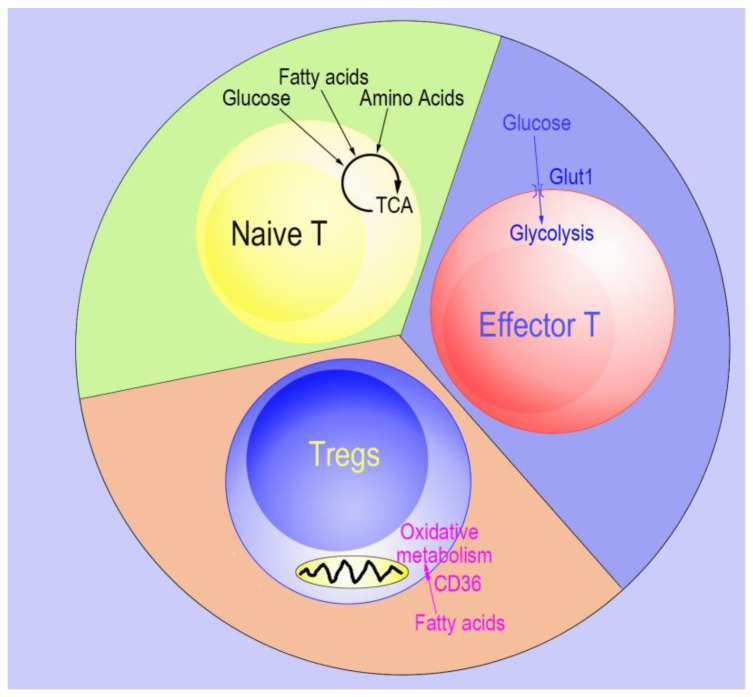
Model of energy usage by naïve T, effector T, and Regulatory T cells (Tregs). Naïve T cells use glucose, fatty acids, and amino acids as their energy source. Effector T cells have higher energy efficiency and use glucose as their primary energy source. In contrast, the glucose transporter 1 is absent in Tregs and Tregs use fatty-acid oxidation (FAO) as their main energy source.

**Figure 2 cells-08-00188-f002:**
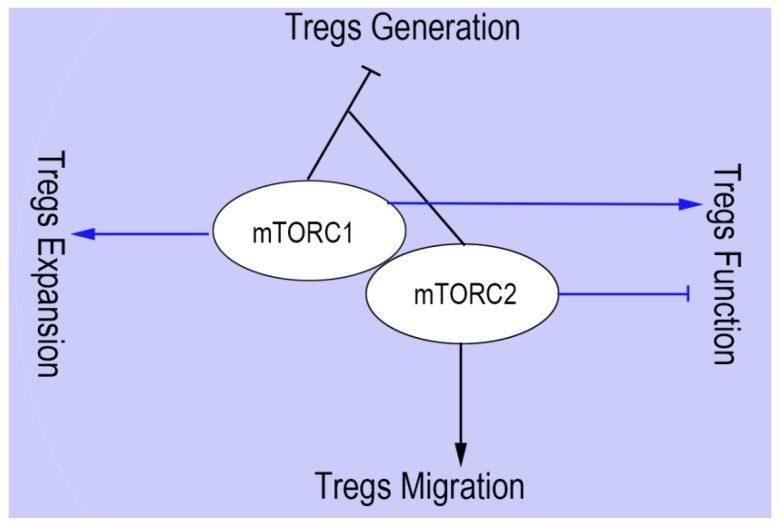
The roles of mTORC1 and mTORC2 on Tregs generation, expansion, function, and migration. The absence of mTOR signaling dramatically increase Tregs generation, while deleting either mTORC1 or mTORC2 signaling does not lead to the upregulation of Foxp3+ Tregs. mTORC1 and mTORC2 play opposite roles in Tregs function, the absence of main component Raptor of mTORC1 limits Tregs function, and lack of mTORC2 increases Tregs function via promoting the activity of mTORC1. mTORC2 promotes the migration of Tregs to inflammatory sites. However, the effects of mTORC1 on the Tregs migration remain unclear. mTOR signaling is essential for Tregs expansion. Consequently, Slc3a2-deficient Tregs have an impaired mTORC1 pathway and show lower proliferation ability. However, the role of mTORC2 on Tregs expansion remains unclear.

**Figure 3 cells-08-00188-f003:**
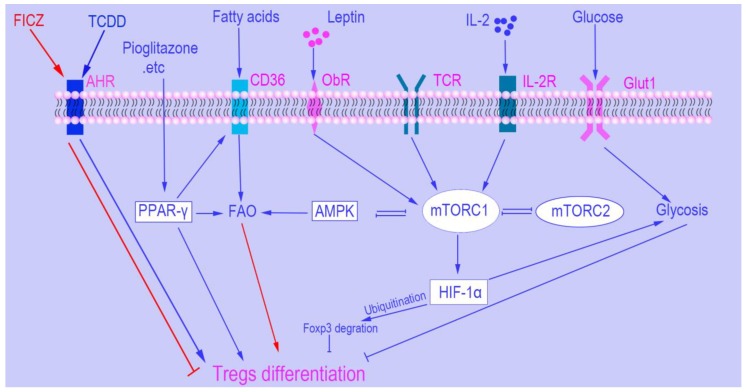
Metabolic and mTOR signaling control of Regulatory T cells’ differentiation. Different ligand binding with aryl hydrocarbon receptor (AHR) leads to dissimilar outcome in Tregs induction. For example, 6-formylindolo [3,2-b] carbazole (FICZ) binding with AHR inhibits Tregs generation while 2,3,7,8-tetrachlorodibenzo-*p*-dioxin (TCDD) promotes Tregs generation. Peroxisome proliferator-activated receptor-γ (PPAR-γ) increase Tregs differentiation via regulating the balance between fatty-acid oxidation (FAO) and glycometabolism. AMP-activated protein kinase (AMPK) is an important metabolic checkpoint in Tregs differentiation, and there is controversy regarding the role of the AMPK in Tregs differentiation. mTOR signaling is the most important metabolic regulator of Tregs. If mTOR is absent in naïve CD4+ T cells, it can dramatically increase Tregs differentiation even under normal activating conditions. HIF1α does not directly regulate the expression of Foxp3. However, it promotes the degradation of Foxp3 protein by ubiquitination.

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
