# Peer review of "Cellular Metabolic Regulation in the Differentiation and Function of Regulatory T Cells"

_cells, 2019, doi:10.3390/cells8020188_

Round 1

Reviewer 1 Report

The manuscript “Cellular metabolic regulation in the differentiation and function of regulatory T cells” by Ye Chen et al. is a brief overview on the relationship between metabolic pathways and Treg cell function and differentiation. This is a timely relevant topic that is generating an increasing amount of literature. Understanding the metabolically-linked control of Treg cell function and differentiation may indeed enhance the prospects for new therapeutic approaches in many immune-related conditions.

 However, two major concerns diminish this reviewer’s enthusiasm for the manuscript:

                1.- A concise review as this should be very precise with the words used. Unfortunately, the manuscript demands a serious editing. As it is, some parts are very difficult to follow through, the message is confusing and often it gets lost.

                2.- The central theme of the manuscript, the metabolism in Treg cells, is not properly structured. The authors have decided to divide the manuscript in 8 sections. Aside sections 1 (overview) and 8 (concluding remarks), there are no links among the rest of the sections. As a reader, you look for a congruent rationale and the development of a message while reading. Why not to focus on mTOR and the mTOR pathway? There is literature about the interplay between mTOR and AMPK, HIF1a, leptin, ….  I am also missing some comments regarding the IL-2 pathway (and the cross-talk with mTOR) and autophagy (and the cross-talk with mTOR), two pathways with well-established metabolic links in Treg cells.

Author Response

Response to Reviewer 1 Comments

The manuscript “Cellular metabolic regulation in the differentiation and function of regulatory T cells” by Ye Chen et al. is a brief overview on the relationship between metabolic pathways and Treg cell function and differentiation. This is a timely relevant topic that is generating an increasing amount of literature. Understanding the metabolically-linked control of Treg cell function and differentiation may indeed enhance the prospects for new therapeutic approaches in many immune-related conditions.

Response: We greatly appreciate review one for your supportive comments.

 However, two major concerns diminish this reviewer’s enthusiasm for the manuscript:

                1.- A concise review as this should be very precise with the words used. Unfortunately, the manuscript demands a serious editing. As it is, some parts are very difficult to follow through, the message is confusing and often it gets lost.

Response: We understand reviewer one’s concern and have carefully edited it accordingly. Jacob Colello, an Immunologist and Biologist has carefully edited English grammar.

                2.- The central theme of the manuscript, the metabolism in Treg cells, is not properly structured. The authors have decided to divide the manuscript in 8 sections. Aside sections 1 (overview) and 8 (concluding remarks), there are no links among the rest of the sections. As a reader, you look for a congruent rationale and the development of a message while reading. Why not to focus on mTOR and the mTOR pathway? There is literature about the interplay between mTOR and AMPK, HIF1a, leptin, ….  I am also missing some comments regarding the IL-2 pathway (and the cross-talk with mTOR) and autophagy (and the cross-talk with mTOR), two pathways with well-established metabolic links in Treg cells.

Response: Reviewer one’s comments are highly constructive. We have re-structured the MS paper and mainly focused on the mTOR related molecules now. Additionally, we also added the IL-2 pathway in Section 3.6 and discussed the autophagy in Section 1.

Reviewer 2 Report

The authors summarized recent discoveries on the metabolic regulation of Tregs. This is an area under intense study. So I would suggest the authors cite the most recent works. For example, there is a debate about Treg reliance of FAO with a recent paper Raud B, et al, Cell Metabolism 2018. Similarly, AMPK mediated signaling may not significantly contributes to Treg, because AMPK deficient Tregs appear to be normal (Yang K, et al, Nature 2017). The function of PPARg in Tregs and metabolic diseases were explored in recent papers Bapat SP, et al, Nature 2015 and Cipolletta D, et al, Nature 2012. The papers had different conclusions and they should be discussed. 

Some editing may help. A few sentences are difficult to understand. For example, line 57-58, 112-115.

Author Response

Response to Reviewer 2 Comments

The authors summarized recent discoveries on the metabolic regulation of Tregs. This is an area under intense study. So I would suggest the authors cite the most recent works. For example, there is a debate about Treg reliance of FAO with a recent paper Raud B, et al, Cell Metabolism 2018.

Response: We thank reviewer two for your supportive comments and have added the controversial opinion in Section 1 with updated literatures.

Similarly, AMPK mediated signaling may not significantly contributes to Treg, because AMPK deficient Tregs appear to be normal (Yang K, et al, Nature 2017).

Response: Thanks for your important advice, we have discussed this article and other related papers in Section 3.2 in revised MS now.

The function of PPARg in Tregs and metabolic diseases were explored in recent papers Bapat SP, et al, Nature 2015 and Cipolletta D, et al, Nature 2012. The papers had different conclusions and they should be discussed. 

Response: This suggestion has been taken and the discussion on these two articles have been added to revised MS now.

Some editing may help. A few sentences are difficult to understand. For example, line 57-58, 112-115.

Response: A native English speaker with Immunology/Biology background has contributed to grammar edit. 

Reviewer 3 Report

The review "Cellular metabolic regulation in the differentiation and function of regulatory T cells" (Chen Y. et al.) focuses primarily on the recent research that investigate the impact of distinct cellular metabolic pathways on CD4+CD25+Foxp3+ Treg cells development, trafficking, and function.  It is emphasized that Treg differentiation depends more on fatty acid oxidation as energy sources, while effector T cells display more glycolytic metabolic demands, as well as that mTOR signaling pathway is a major player in T cell fate determination, owing to its ability to integrate input signals from major intracellular and extracellular cues, such as growth factors, energy status, oxygen, stress, and nutrients. Besides, the evidence linking the mTOR pathway with Treg differentiation, Treg function, Treg expansion and Treg migration have been presented. In addition, the regulatory effects of other elements with high impact on cellular energy metabolism, such as hypoxia-inducible factor (HIF1α), AMP-activated protein kinase, leptin, peroxisome proliferator-activated receptors, and the aryl hydrocarbon receptor on Treg cells generation and function are highlighted. 

The data from collected references clearly showed that the development and immunosuppressive functions of CD4+CD25+Foxp3+ Treg cells are under the control of intracellular pathways that regulate the glycolytic and lipid oxidative metabolic programs during T-cell differentiation. The review is interesting and written by experts in the field. Owing to the crucial function of Tregs in maintenance of immune tolerance and prevention of autoimmune and inflammatory diseases it might be of boarder clinical interest. I would recommend accepting it after minor essential revision.

Minor Essential Revisions

For better understanding of the presented data it would be helpful to add the following:

In section 2. mTOR

the      short description of structures and functions of two mTOR complexes that      are differently sensitive to rapamycin

the      short description of mTORC1 signaling cascade and its link with activating      upstream signaling molecules, such as the phosphoinositide-3-kinase      (PI3K)-AKT pathway and downstream metabolic pathways, particularly those      related with metabolism of lipids

In Figure 1:

·         Short description of metabolic pathways used by shown T cell subsets for the energy usage. Highlight the absence of Glut1 in Treg cells.

      - Change the name of "effect" T cells to "effector" T cells

       In Figure 2.

·         Short description of shown metabolic pathways and their relationships 

Line 162  It should be better explained what is presented in Box 1

In the whole text:

·         the names of numerous abbreviations should be explained at the first appearance or shown in separate section.

Author Response

Response to Reviewer 3 Comments

The review "Cellular metabolic regulation in the differentiation and function of regulatory T cells" (Chen Y. et al.) focuses primarily on the recent research that investigate the impact of distinct cellular metabolic pathways on CD4+CD25+Foxp3+ Treg cells development, trafficking, and function.  It is emphasized that Treg differentiation depends more on fatty acid oxidation as energy sources, while effector T cells display more glycolytic metabolic demands, as well as that mTOR signaling pathway is a major player in T cell fate determination, owing to its ability to integrate input signals from major intracellular and extracellular cues, such as growth factors, energy status, oxygen, stress, and nutrients. Besides, the evidence linking the mTOR pathway with Treg differentiation, Treg function, Treg expansion and Treg migration have been presented. In addition, the regulatory effects of other elements with high impact on cellular energy metabolism, such as hypoxia-inducible factor (HIF1α), AMP-activated protein kinase, leptin, peroxisome proliferator-activated receptors, and the aryl hydrocarbon receptor on Treg cells generation and function are highlighted. 

The data from collected references clearly showed that the development and immunosuppressive functions of CD4+CD25+Foxp3+ Treg cells are under the control of intracellular pathways that regulate the glycolytic and lipid oxidative metabolic programs during T-cell differentiation. The review is interesting and written by experts in the field. Owing to the crucial function of Tregs in maintenance of immune tolerance and prevention of autoimmune and inflammatory diseases it might be of boarder clinical interest. I would recommend accepting it after minor essential revision.

 Response: we highly appreciate reviewer three’s supportive comments.

Minor Essential Revisions

For better understanding of the presented data it would be helpful to add the following:

In section 2. mTOR

the      short description of structures and functions of two mTOR complexes that      are differently sensitive to rapamycin

the      short description of mTORC1 signaling cascade and its link with activating      upstream signaling molecules, such as the phosphoinositide-3-kinase      (PI3K)-AKT pathway and downstream metabolic pathways, particularly those      related with metabolism of lipids

Response: These suggestions are crucial and we have added more details of the mTOR signaling in revised MS now.

In Figure 1:

·         Short description of metabolic pathways used by shown T cell subsets for the energy usage. Highlight the absence of Glut1 in Treg cells.

      - Change the name of "effect" T cells to "effector" T cells

Response: These suggests have been taken and a short description of Figure 1 has been added now. We also modified the typo.

       In Figure 2.

·         Short description of shown metabolic pathways and their relationships 

 Line 162  It should be better explained what is presented in Box 1

Response:  We have added some description of Fig 3 (previous Fig 2) and some explanations of Box 1 in the related parts in revised MS now.

In the whole text:

·         the names of numerous abbreviations should be explained at the first appearance or shown in separate section.

Response: Thanks for your valuable comments, we have updated all the explanations of abbreviation.

Reviewer 4 Report

In the Review: "Cellular metabolic regulation in the differentiation and function of regulatory T cells" the authors addressed an interesting topic on how differential aspects of Treg biology are regulated by intracellular metabolic processes. The manuscript is full of signaling molecules and other factors that makes it hard to distinguish which of them are important and which of them are rather of minor importance for Treg biology. Moreover, some of the molecules are introduced by their full names some of them not. In general, abbreviations should be introduced when the molecule is mentioned the first time and afterwards used consistently throughout the text. Figure 1 is not very informative. This figure can be replaced by a scheme showing the different effects of mTORC1 and mTORC2 on Treg generation, expansion and migration. Alternatively, a new figure can be added. Then, the reviewer does not understand what is meant by: "...high doses of IL-2 and rapamycin ...activator of mTOR kinase pathway and a specific inhibitor of mTOR kinase (lines 93-95)". Do IL-2 and rapamycin function in a negative feedback loop mechanism? Figure 2 should also include Ahr. In Box 1, the reviewer would recommend to replace "stable states" by "steady states".      

Author Response

Response to Reviewer 4 Comments

In the Review: "Cellular metabolic regulation in the differentiation and function of regulatory T cells" the authors addressed an interesting topic on how differential aspects of Treg biology are regulated by intracellular metabolic processes. The manuscript is full of signaling molecules and other factors that makes it hard to distinguish which of them are important and which of them are rather of minor importance for Treg biology. Moreover, some of the molecules are introduced by their full names some of them not. In general, abbreviations should be introduced when the molecule is mentioned the first time and afterwards used consistently throughout the text.

Response: We really appreciate your precious comments, we have re-structured this review and updated all the explanation of abbreviation.

 Figure 1 is not very informative. This figure can be replaced by a scheme showing the different effects of mTORC1 and mTORC2 on Treg generation, expansion and migration. Alternatively, a new figure can be added. Then, the reviewer does not understand what is meant by: "...high doses of IL-2 and rapamycin ...activator of mTOR kinase pathway and a specific inhibitor of mTOR kinase (lines 93-95)". Do IL-2 and rapamycin function in a negative feedback loop mechanism?

Response: Thanks for your comments, we have added one figure to describe the effects of mTORC1 and mTORC2 on Treg generation, expansion and migration. IL-2 is an activator of mTOR and rapamycin inhibits the activity of mTOR, while we still need to add high does IL-2 and rapamycin to expand the Treg cells in vitro. Nonetheless, whether IL-2 and rapamycin function in a negative feedback loop mechanism still remains unclear so far.

Figure 2 should also include Ahr. In Box 1, the reviewer would recommend to replace "stable states" by "steady states".    

Response: We thank the reviewer for this valuable suggestion. We have added Ahr. to Figure 3 (we have `added a new figure before previous Figure 2) and changed “Stable states” with “steady states” now.

Round 2

Reviewer 1 Report

This reviewer recognizes the authors’ effort to improve the structure and content of their manuscript “Cellular metabolic regulation in the differentiation and function of regulatory T cells”. However, there still remain issues in the revised form that makes for not a comfortable lecture and should be addressed. Some of them are simple edits that a more careful screening should have easily avoided. The comments are orderly indicated by numerical lines to make it easy to navigate.

·         Line 28: “conventional T (Tcon) cells and naïve T cells“. In the context of Tregs, conventional T cells are the non-Treg T cells, which include naive T cells. “Effector conventional T cells” would likely be a more appropriate term.

·         Lines 33-34 “In addition, the expression of Foxp3 in Treg cells inhibits Myc expression and reduces glycolysis , which can be suitable for Treg cells in low glucose condition.” Please add a reference supporting this statement [for instance,  ref#16: Angelin A et al  Cell Metab. 6;25(6):1282-1293, 2017]

·         Lines 39-43 “Long chain fatty (LC) acid promotes Teff cell differentiation and inhibits Treg differentiation [19], while Raun et al. presented the controversol opinion that long chain-FAO (LC-FAO) does not affect the generation of Tregs[20]. Carnitine palmitoyltransferase 1a(Cpt1a) is an important regulator in FAO of T cells and is not necessary for the generation and function of CD4+ Tregs [20]“. Kind of intricate paragraph that seems to support LC as inhibitors of Treg differentiation (19), but not (20). Could you please rephrase the paragraph?

·         Line 56: mTOR was already mentioned in line #38. The abbreviation should be noted then in #38.

·         Lines 59-60: “It is recongnized that mTORC1 is more sensitive to rapamycin than mTORC2 [24],while it is almost the same sensitive in Naïve CD4+ T cells[25]“. Please rephrase the paragraph. Besides the misspelling of “recongnized”, the sentence “while it is almost the same sensitive in Naïve CD4+ T cells” is difficult to understand.

·         Line 61: “the modulation of metabolic regulation” the modulation of any regulation sounds redundant. Please, use other terms.

·         Line 76: “activation of PI3K leads to activate the serine-threonise kinase Atk”. The authors likely meant “AKT” and “serine-threonine”. For consistency, please identify the acronym AKT.

·         Line 101:  “Instead, When mTOR…”.  You must never use a capital letter after a comma, except when it is a 'Proper Noun'.

·         Lines 117-118: “Spontaneous effector T cell activation is driven by Treg-specific deletion of mTOR”. However in lines 101-102 the authors indicate that: “When mTORC2 and mTORC1 were both mutually absent, naïve CD4+ T cells were differentiated into Foxp3+ Treg cells even under normally activating conditions[32]”. Could you please clarify this apparent inconsistency?

·         Line 119: Could you please clarify the term “effector Tregs”? The authors never mentioned the potential differences among different subsets of Treg cells (namely, thymic-derived, in vivo peripherally derived and in vitro derived Tregs).

·         Line 124: “Inhibition of mTOR enhances Treg expansion in vitro”. In contrast, in lines 119-120: “Toll-like receptor (TLR) signals up-regulated Treg cell proliferation through mTORC1 signaling pathway, glucose transporter 1 (Glut1) upregulation and glycolysis”. Could you please clarify this apparent inconsistency? How the rapamycin-induced mTOR inhibition can enhance the expansion and the mTORC1 signaling also?  

·         Line 141: The term “Movement” is not the most appropriate when referring to cell trafficking or migration.

·         Line 143: “… culminated in stimulation of the enzyme glucokinase (GCK). Where after, associating…”. Where after (?).

·          Line 184: please clarify the term “bidirectional” as referred to the mTOR role in Treg cells.

·         Line 185. “(…) induction and function, it is necessary to find more…”. Please, separate the sentences with a period (.) instead of a comma (,).

·          Line 186. “ (…) the homestasis of immune balance,next we will discus (…)”.Please, add a period (.) after “balance”. Change homestasis to “homeostasis”.

·         Line 189: “ (…) (HIF1α) is an imperative protein for sensing…”. The term “imperative” feels out-of-place to adjectivize a protein.

·         Lines 191-193. The sentence “mTOR pathway. mTOR is required to determine cell  fate through regulating the metabolic responses of T cells. mTOR is stimulated by T cell activation to promote glycolysis and diminish lipid oxidation” is redundant since the same information has been showed before in the text.

·         Line 194: “ (…) production of Treg cells that occurs through inhibition of HIF1α, which is mediated by mTOR[44]”. The expression and activity of HIF1a depends on functional mTOR. The sentence, as written, may draw the wrong assumption that the inhibition of HF1a is mediated by the activation mTOR. Please rephrase it.

·         Lines 195-198. The whole paragraph is poorly written with short disjointed sentences. The information is interesting but the argument needs to be clearer and with cohesion among the parts. Please use verbal tense consistency in the same sentence (“HIF1α promotes Treg induction and protected mice”)

·         Lines 213-214: What is “metabolic kinase”?

·         Line 215: “ (…) AMPK activation by Ca2+-dependent protein.” One? Which one?

·         Lines 220-221: “ (…) Treg cells rely mostly on FAO metabolism, while naïve T cells can be prompted to differentiate toward Treg cells (…)”. This is a poorly constructed sentence. Please, make it clearer. Do you refer to the distinct effects promoted by FAO in Tregs and naïve T cells?

·         Line 222: What is doing the sentence “While some other groups found some different results.” hanging there?

·         Line 223. It should change the comma (,) by a period (.), as the next sentence is completely different (!).

·         Line 225: “…while it is independent on AMPK[62,63]”. Two lines above: “Liver kinase B1(LKB1) is a best-studied (?) upstream kinase of AMPK”. Please comment this apparent contradiction.

·         Lines 225-226: “Moreover, a recent paper demonstrated that Tregs differentiation is independent on the AMPK-Driven LC-FAO[20]”. Is there a link between this sentence and the paragraph of lines 39-43? Could it be integrated there?

·         The section 3.2 (lines 212-226) needs a complete rewriting.

·          Line 233: The use of “Thus” is incorrect. The following sentence is not a consequence of the previous.

·         Lines 238-241: “PPARs play an important  role in peroxisomal mediated β-oxidation of FAO, when PPARs are activated with specific ligands, conformational changes occur, resulting in heterodimerization with retinoid X receptor (RXRα), which binds to promoter regions of target genes involved in FAO [71,72].” Please, use correct punctuation marks.

·         Line 248: “(…) proved to uniquely express PPARγ,which are specifically recruited in adipose tissue.” PPARγ is not recruited to adipose tissue. If the authors refer to Treg cells, please rephrase the sentence accordingly.

·         Lines 252-254: “(…) the absence of PPARγ in  VAT Treg can not perturb the frequency in aged mice compared to control mice, and these Treg cluster less closely with splenic Tregs than fat Tcon cells[78]”. Please, rephrase the sentences and add a comment to underscore the significance of these findings.

·         Line 286: “We have confirmed, along with several groups, that IL-2 was necessary for (…)”. Please use the correct verbal tense (IL-2 is still necessary to induce and expand Treg cells).

·         Line 287: Does nTregs mean thymic-derived Tregs? Is there a difference among the different subpopulations of Tregs?

·         Lines 294-295: “there is complex interplay between intrinsic metabolic pathways and Tregs”. Please rephrase the sentence: It is conceptually impossible the communication (interplay) between metabolic pathways and Tregs as the former are part of the later. The interplay may occur between signaling and metabolic networks IN Tregs or between conventional and regulatory T cells in vivo or in vitro.  It could also be that the precise role of the metabolic pathways in the regulation of Treg cell differentiation, expansion and function is complex.

Author Response

Response to Reviewer 1 Comments

This reviewer recognizes the authors’ effort to improve the structure and content of their manuscript “Cellular metabolic regulation in the differentiation and function of regulatory T cells”. However, there still remain issues in the revised form that makes for not a comfortable lecture and should be addressed. Some of them are simple edits that a more careful screening should have easily avoided. The comments are orderly indicated by numerical lines to make it easy to navigate.

Response: We really appreciate your constructive comments.

·         Line 28: “conventional T (Tcon) cells and naïve T cells“. In the context of Tregs, conventional T cells are the non-Treg T cells, which include naive T cells. “Effector conventional T cells” would likely be a more appropriate term.

Response: Thanks for your precise word, we have changed the term now.

·         Lines 33-34 “In addition, the expression of Foxp3 in Treg cells inhibits Myc expression and reduces glycolysis, which can be suitable for Treg cells in low glucose condition.” Please add a reference supporting this statement [for instance,  ref#16: Angelin A et al  Cell Metab. 6;25(6):1282-1293, 2017]

Response: The reference has been added now.

·         Lines 39-43 “Long chain fatty (LC) acid promotes Teff cell differentiation and inhibits Treg differentiation [19], while Raun et al. presented the controversial opinion that long chain-FAO (LC-FAO) does not affect the generation of Tregs[20]. Carnitine palmitoyltransferase 1a(Cpt1a) is an important regulator in FAO of T cells and is not necessary for the generation and function of CD4+ Tregs [20]“. Kind of intricate paragraph that seems to support LC as inhibitors of Treg differentiation (19), but not (20). Could you please rephrase the paragraph?

Response: Thanks for your suggestion, we have re-edited this paragraph in the revised MS.

·         Line 56: mTOR was already mentioned in line #38. The abbreviation should be noted then in #38.

Response: Agree, we have updated the abbreviation of mTOR.

·         Lines 59-60: “It is recognized that mTORC1 is more sensitive to rapamycin than mTORC2 [24], while it is almost the same sensitive in Naïve CD4+ T cells[25]“. Please rephrase the paragraph. Besides the misspelling of “recognized”, the sentence “while it is almost the same sensitive in Naïve CD4+ T cells” is difficult to understand.

Response: Thanks for your valuable comments, we have rewritten this sentence.

·         Line 61: “the modulation of metabolic regulation” the modulation of any regulation sounds redundant. Please, use other terms.

Response: Thanks for your advice, we have changed the terms.

·         Line 76: “activation of PI3K leads to activate the serine-threonise kinase Atk”. The authors likely meant “AKT” and “serine-threonine”. For consistency, please identify the acronym AKT.

Response: We have updated the meaning of AKT now.

·         Line 101:  “Instead, When mTOR…”.  You must never use a capital letter after a comma, except when it is a 'Proper Noun'.

Response: Agree, we have modified the typo.

·         Lines 117-118: “Spontaneous effector T cell activation is driven by Treg-specific deletion of mTOR”. However in lines 101-102 the authors indicate that: “When mTORC2 and mTORC1 were both mutually absent, naïve CD4+ T cells were differentiated into Foxp3+ Treg cells even under normally activating conditions[32]”. Could you please clarify this apparent inconsistency?

Response: Thanks for your comments, we have rewritten this sentence.

·         Line 119: Could you please clarify the term “effector Tregs”? The authors never mentioned the potential differences among different subsets of Treg cells (namely, thymic-derived, in vivo peripherally derived and in vitro derived Tregs).

Response: The term “effector Tregs” is from the cited paper (34), the author referred to the thymic-derived Tregs in vivo. In order to avoid misunderstanding, we have deleted the word “effector” and added the description of different Tregs subsets in Section 1.

·         Line 124: “Inhibition of mTOR enhances Treg expansion in vitro”. In contrast, in lines 119-120: “Toll-like receptor (TLR) signals up-regulated Treg cell proliferation through mTORC1 signaling pathway, glucose transporter 1 (Glut1) upregulation and glycolysis”. Could you please clarify this apparent inconsistency? How the rapamycin-induced mTOR inhibition can enhance the expansion and the mTORC1 signaling also?  

Response: Thanks for your valuable comments, we have modified the sentence. We believe rapamycin inhibits mTOR signal to promote Treg differentiation. However, if appropriate IL-2 exists, the combination of both rapamycin and IL-2 facilitates Treg expansion even mTORC1 is inhibited.

·         Line 141: The term “Movement” is not the most appropriate when referring to cell trafficking or migration.

Response: We have replaced the word.

·         Line 143: “… culminated in stimulation of the enzyme glucokinase (GCK). Where after, associating…”. Where after (?).

Response: The sentence has been modified now.

·          Line 184: please clarify the term “bidirectional” as referred to the mTOR role in Treg cells.

Response: One sentence is added to clarify this term in the revised MS.

·         Line 185. “(…) induction and function, it is necessary to find more…”. Please, separate the sentences with a period (.) instead of a comma (,).

Response: The change is made now.

·          Line 186. “ (…) the homestasis of immune balance, next we will discus (…)”.Please, add a period (.) after “balance”. Change homestasis to “homeostasis”.

Response: We have separated the sentence and modified the typo.

·         Line 189: “ (…) (HIF1α) is an imperative protein for sensing…”. The term “imperative” feels out-of-place to adjectivize a protein.

Response: Agree, we have changed the word.

·         Lines 191-193. The sentence “mTOR pathway. mTOR is required to determine cell fate through regulating the metabolic responses of T cells. mTOR is stimulated by T cell activation to promote glycolysis and diminish lipid oxidation” is redundant since the same information has been showed before in the text.

ResponseAgree. We have deleted this information here.

·         Line 194: “ (…) production of Treg cells that occurs through inhibition of HIF1α, which is mediated by mTOR[44]”. The expression and activity of HIF1a depends on functional mTOR. The sentence, as written, may draw the wrong assumption that the inhibition of HF1a is mediated by the activation mTOR. Please rephrase it.

Response: Thanks for your suggestion, we have rewritten this sentence.

·         Lines 195-198. The whole paragraph is poorly written with short disjointed sentences. The information is interesting but the argument needs to be clearer and with cohesion among the parts. Please use verbal tense consistency in the same sentence (“HIF1α promotes Treg induction and protected mice”)

Response: We really appreciate your comments. We have rewritten this paragraph now.

·         Lines 213-214: What is “metabolic kinase”?

Response:  We have replaced the term.

·         Line 215: “ (…) AMPK activation by Ca2+-dependent protein.” One? Which one?

Response: AMPK can be activated by Ca2+ dependent protein (CaMK).

·         Lines 220-221: “ (…) Treg cells rely mostly on FAO metabolism, while naïve T cells can be prompted to differentiate toward Treg cells (…)”. This is a poorly constructed sentence. Please, make it clearer. Do you refer to the distinct effects promoted by FAO in Tregs and naïve T cells?

Response: We have rewritten these sentences.

·         Line 222: What is doing the sentence “While some other groups found some different results.” hanging there?

Response: Thanks for your concern, we have deleted this sentence.

·         Line 223. It should change the comma (,) by a period (.), as the next sentence is completely different (!).

Response: Thanks for your suggestion, we have edited this now.

·         Line 225: “…while it is independent on AMPK[62,63]”. Two lines above: “Liver kinase B1(LKB1) is a best-studied (?) upstream kinase of AMPK”. Please comment this apparent contradiction.

Response: Agree. We have added more information to clarify this contradiction.

·         Lines 225-226: “Moreover, a recent paper demonstrated that Tregs differentiation is independent on the AMPK-Driven LC-FAO[20]”. Is there a link between this sentence and the paragraph of lines 39-43? Could it be integrated there?

Response: Thanks for your important suggestion, in this section, we mainly discuses AMPK here and this paper indicates that AMPK may not affect the Treg differentiation.

·         The section 3.2 (lines 212-226) needs a complete rewriting.

Response: We have rewritten this section, hopefully, it is much better now.

·          Line 233: The use of “Thus” is incorrect. The following sentence is not a consequence of the previous.

Response: Thanks. We have replaced this word now.

·         Lines 238-241: “PPARs play an important role in peroxisomal mediated β-oxidation of FAO, when PPARs are activated with specific ligands, conformational changes occur, resulting in heterodimerization with retinoid X receptor (RXRα), which binds to promoter regions of target genes involved in FAO [71,72].” Please, use correct punctuation marks.

Response: Change is made now.

·         Line 248: “(…) proved to uniquely express PPARγ, which are specifically recruited in adipose tissue.” PPARγ is not recruited to adipose tissue. If the authors refer to Treg cells, please rephrase the sentence accordingly.

Response: Thanks for your suggestion, we have modified this sentence.

·         Lines 252-254: “(…) the absence of PPARγ in  VAT Treg can not perturb the frequency in aged mice compared to control mice, and these Treg cluster less closely with splenic Tregs than fat Tcon cells[78]”. Please, rephrase the sentences and add a comment to underscore the significance of these findings.

Response: We have rephrased the sentences and added a comment on its importance.

·         Line 286: “We have confirmed, along with several groups, that IL-2 was necessary for (…)”. Please use the correct verbal tense (IL-2 is still necessary to induce and expand Treg cells).

Response: Agree.

·         Line 287: Does nTregs mean thymic-derived Tregs? Is there a difference among the different subpopulations of Tregs?

Response: Yes, nTregs are derived from thymus (also named tTreg). Other subsets include pTregs that are generated extrathymically at peripheral sites and iTregs that are developed or induced ex vivo in the presence of TCR signal with transforming growth factor beta and IL-2.

·         Lines 294-295: “there is complex interplay between intrinsic metabolic pathways and Tregs”. Please rephrase the sentence: It is conceptually impossible the communication (interplay) between metabolic pathways and Tregs as the former are part of the later. The interplay may occur between signaling and metabolic networks IN Tregs or between conventional and regulatory T cells in vivo or in vitro.  It could also be that the precise role of the metabolic pathways in the regulation of Treg cell differentiation, expansion and function is complex.

Response:  Agree. The change has been made now.

Reviewer 4 Report

In their revised manuscript the authors provided additional information and included/revised figures. Unfortunately, the English language is still of poor quality and need another round of editing. The text contains several grammatical and spelling errors. Additionally, the authors still use different abbreviations for Treg cells (Treg or Treg cells or Tregs) and effector T cells, did not introduce the abbreviation when used the first time (e.g. FAO, PI3K) and used words incorrectly (e.g. autophage instead of autophagy). 

Author Response

Response to Reviewer 4 Comments

In their revised manuscript the authors provided additional information and included/revised figures. Unfortunately, the English language is still of poor quality and need another round of editing. The text contains several grammatical and spelling errors.

Response: MS R2 has been further edited by Dr. Wael Jarjour, a chief of Division of Rheumatology and Immunology at OSU College of Medicine to avoid possible errors.

Additionally, the authors still use different abbreviations for Treg cells (Treg or Treg cells or Tregs) and effector T cells, did not introduce the abbreviation when used the first time (e.g. FAO, PI3K) and used words incorrectly (e.g. autophage instead of autophagy). 

Response: We appreciate for picking up. We have modified the different abbreviations and added the abbreviation in the first place. We have also changed the incorrect words now.